# Characterization of Reconstructed Ancestral Proteins Suggests a Change in Temperature of the Ancient Biosphere

**DOI:** 10.3390/life7030033

**Published:** 2017-08-06

**Authors:** Satoshi Akanuma

**Affiliations:** Faculty of Human Sciences, Waseda University, 2-579-15 Mikajima, Tokorozawa, Saitama 359-1192, Japan; akanuma@waseda.jp; Tel.: +81-4-2947-6727; Fax: +81-4-2947-6811

**Keywords:** ancestral sequence reconstruction, ancient biosphere, last universal common ancestor, phylogenetic analysis, Precambrian, thermophilicity

## Abstract

Understanding the evolution of ancestral life, and especially the ability of some organisms to flourish in the variable environments experienced in Earth’s early biosphere, requires knowledge of the characteristics and the environment of these ancestral organisms. Information about early life and environmental conditions has been obtained from fossil records and geological surveys. Recent advances in phylogenetic analysis, and an increasing number of protein sequences available in public databases, have made it possible to infer ancestral protein sequences possessed by ancient organisms. However, the in silico studies that assess the ancestral base content of ribosomal RNAs, the frequency of each amino acid in ancestral proteins, and estimate the environmental temperatures of ancient organisms, show conflicting results. The characterization of ancestral proteins reconstructed in vitro suggests that ancient organisms had very thermally stable proteins, and therefore were thermophilic or hyperthermophilic. Experimental data supports the idea that only thermophilic ancestors survived the catastrophic increase in temperature of the biosphere that was likely associated with meteorite impacts during the early history of Earth. In addition, by expanding the timescale and including more ancestral proteins for reconstruction, it appears as though the Earth’s surface temperature gradually decreased over time, from Archean to present.

## 1. Introduction

There is still limited understanding of ancestral life on Earth, and the environment in which it evolved. Information about early life and the biosphere has often been obtained from fossil records and geological surveys [1,2]. In 1993, Schopf discovered fossilized stromatolite-like structures in the Apex chert from 3.5 gigayears ago (Gya) [1]. Recently, Nutman et al. reported evidence for ancient life obtained from a newly exposed outcrop of 3.7 Gya metacarbonate rocks in the Isua supracrustal belt [3]. Dodd et al. also reported putative microfossils of microorganisms that are possibly 4.3 Gya old in ferruginous sedimentary rocks from the Nuvvuagittuq belt in Quebec, Canada [4]. Evidence for the existence of methanogens and microbial sulfate reduction at 3.5 Gya [5,6] and the emergence of life at 3.8–4.1 Gya have also been reported [7,8].

A growing amount of genomic data available in public databases provide the necessary resource for the study of molecular phylogeny. By comparing a large number of homologous gene or protein sequences, we can now infer the sequences of genes and proteins that were possessed by ancestral organisms [9,10,11,12]. In addition, we can also synthesize the inferred nucleotide and amino acid sequences [9,10,11,13]. Since the physical properties of extant proteins are well adapted to their hosts’ environment, the same must have been true for primitive proteins that existed earlier than 3.5 Gya. Therefore, the nature of ancestral organisms and their environments can be inferred by the reconstruction and characterization of their proteins [10,12,14,15]. In this review, I will discuss the environmental temperatures experienced by ancient organisms that existed during the Precambrian era, as inferred from amino acid sequences of ancient proteins reconstructed by comparing modern homologous sequences.

## 2. Early Studies on the Environmental Temperatures of Ancestral Life

Although there has been a long-running debate about the environment of early life, no consensus has yet been obtained (Table 1). In particular, there has been intense debate about the environmental temperature of the last universal common ancestor. Although the last universal common ancestor is sometimes called Commonote [16] or *Commonote commonote* [17], I hereafter refer to it as LUCA, the most commonly used term. LUCA is not the oldest life, but rather the most recent common ancestor of all modern life. In the phylogenetic tree built by Furukawa et al. [18] (Figure 1), the left end corresponds to the origin of life. LUCA (indicated as ‘a’) is an intermediate ancestor from the origin of life to modern organisms.

According to a frequently referenced phylogenetic tree based on small subunit ribosomal RNA sequences, branches for hyperthermophilic archaea and bacteria are concentrated near the root of the tree [28,29,30]. Therefore, the archaeal and bacterial common ancestors were both thought to be hyperthermophilic organisms [19,20]. In addition, Occam’s razor suggests that their common ancestor, that is LUCA, was also hyperthermophilic and lived in a hot environment.

However, the theory of hyperthermophilic ancestry has often been criticized. Miller and Lazcano [31] argued that it was not likely that the earliest life was hyperthermophilic, because bio-related materials such as ATP are thermally instable. Indeed, it has been empirically demonstrated that ribose, a backbone of RNA, and its analogs quickly decompose at high temperatures [32]. Doolittle [33] pointed out that it is quite difficult to properly represent the early history of life on a tree. Therefore, an accurate tree cannot be obtained and any implications derived from the tree are hard to prove. Indeed, on a tree representing bacterial phylogeny built by Brochier and Philippe [21], the shortest and deepest branches were not for hyperthermophilic bacteria such as *Thermotogales* and *Aquificales*, but rather for mesophilic species. Therefore, they asserted that the hyperthermophilic bacteria emerged as a result of a secondary adaptation to high temperature.

Galtier et al. [22] thought that inferring the guanine plus cytosine (G + C) content of an ancestral rRNA would provide a powerful method to predict the optimum temperature of the ancestral organism. The G + C content of the stem region of prokaryotic ribosomal RNA (rRNA) and the optimum environmental temperature of the host organism are well correlated; an extant prokaryote with greater G + C content in rRNA often shows a higher optimum environmental temperature. The calculated G + C content of LUCA was not similar to the values found for organisms living at high temperatures. Accordingly, Galtier et al. proposed that LUCA was likely a mesophile. However, Di Giulio reanalyzed the same genome data set using a different computational algorithm, and obtained the contradicting conclusion that LUCA was thermophilic or hyperthermophilic [23,34,35].

Reverse gyrase, an ATP-dependent type I DNA topoisomerase, is possessed by all known hyperthermophilic species, and is therefore thought to be an essential protein for adaptation to very high temperatures [36,37]. Accordingly, the emergence of reverse gyrase might be crucial for the origin of hyperthermophilic organisms. Reverse gyrase consists of topoisomerase and helicase domains, which are evolutionarily independent of each other [38]. It is reasonably assumed that the topoisomerase and helicase families evolved independently in mesophilic or thermophilic organisms prior to the emergence of reverse gyrase [39]. Later, the domains fused to each other, and then were recruited by hyperthermophilic organisms [24]. This argument suggests that hyperthermophiles are descendants of mesophilic or thermophilic organisms. However, it cannot be ruled out that reverse gyrase had evolved prior to the emergence of LUCA. Indeed, a very recent study indicated that a gene for reverse gyrase was included in a 355-gene set that might have been possessed by LUCA [25]. Therefore, the discussion about the origin and evolution of reverse gyrase is compatible with the following scenario: at the time when the universal ancestor lived, a variety of organisms existed in a wide range of temperature environments, and when the surface temperature of Earth drastically increased due to meteorite impacts and various other reasons, only a hyperthermophilic ancestor survived [40].

## 3. Experimental Procedure to Reconstruct an Ancestral Protein Sequence

Since the end of the 20th century, the history of life and proteins has been traced by comparing the amino acid sequences of homologous proteins [15,17,41,42,43,44,45,46,47,48]. Ancestral protein sequences have also been inferred computationally [9,12,49,50]. Figure 2 shows a flowchart for reconstructing an ancestral protein. The first step of the reconstruction is to retrieve homologous protein sequences of the target protein from public databases. The homologous sequences are then aligned to generate a multiple sequence alignment. Our group often uses MAFFT [51] to align a set of homologous sequences, but manually corrects the positions of insertions and gaps if necessary. Next, the alignment and the homologous sequences are used to build a phylogenetic tree. An ancestral amino acid sequence is then computed using the tree topology, the homologous sequences contained in the tree, and either a homogeneous or a non-homogeneous amino acid substitution model. The homogeneous model uses an approximation of constant global amino acid compositions in proteins throughout evolution [52]. In contrast, the non-homogeneous evolution model relaxes this constraint and allows for different global amino acid compositions at different times and for different lineages of the tree [22,53]. The positions of gaps/inserts are given in the ancestral sequence. We often use the program GASP [54] for this purpose. The gene encoding the inferred amino acid sequence is synthesized in vitro, and then expressed in a host organism such as *Escherichia coli*. Finally, the recombinant ancestral protein is purified and characterized experimentally. Theories and procedures of the reconstruction technique are described in greater detail in excellent reviews by Thornton [9], Gaucher et al. [10], and Merkl and Sterner [11].

## 4. Experimentally Testing If Ancestral Organisms Were Thermophiles

Yamagishi and coworkers developed an experimental way to test the thermophilicity of LUCA. They first inferred an ancestral amino acid sequence of a protein that might be possessed by LUCA. Then, one or a few amino acid(s) found in the inferred ancestral sequence were introduced into a protein from a modern thermophilic organism as amino acid substitution(s); then, the thermal stability of the mutant proteins was assessed. Using this method, they constructed mutants for 3-isopropylmalate dehydrogenase (IPMDH) from the hyperthermophile *Sulfolobus tokodaii* [55], isocitrate dehydrogenase from the extremely thermophile *Caldococcus noboribetus* [56], and glycyl-tRNA synthetase [57] and IPMDH [58] from the extremely thermophile *Thermus thermophilus*. From these experiments, they observed that the mutant proteins showed a trend toward greater thermal stability than the wild-type proteins. They asserted that their observations were evidence that LUCA possessed very thermostable proteins, thus supporting the hyperthermophilicity of LUCA. Similar methods also improved the thermal stability of mesophilic proteins [59,60,61]. However, they compared a relatively small number of homologous amino acid sequences to infer the ancestral sequences, and it is not likely that a tree based on a small number of homologous sequences would accurately reflect the phylogeny of all modern life. Therefore, the inferred sequences might not represent the true ancestral sequences. Accordingly, the observed trend for increased thermal stability of mutant proteins may be an artifact of the sequence inference method [62].

Genes encoding entire ancestral amino acid sequences were synthesized to study the evolution of protein function. In 2005, ancestral forms of yeast alcohol dehydrogenase were reconstructed to determine the original function of the protein [42]. Thornton and coworkers intensively investigated changes in ligand specificities of steroid receptors [63,64,65,66]. The same group also used an ancestral sequence reconstruction technique to study the evolutionary process of increased complexity of eukaryotic V-ATPase proton pumps [67]. The experimental reconstruction method was also used to address the history of recruiting 20 genetically coded amino acids [46]. Thus, the reconstruction of an entire ancestral amino acid sequence in vitro is a commonly used technique to understand the histories of proteins and their host organisms.

Gaucher et al. [41] applied this technique to investigate the environmental temperature experienced by early organisms. They reconstructed two ancestral elongation factor-Tu proteins corresponding to the last bacterial common ancestor, and then characterized the optimum temperatures of the GTP hydrolysis activity. The ancestral proteins demonstrated optimum activity at temperatures similar to that of a modern thermophilic elongation factor-Tu, supporting the theory that the last bacterial common ancestor was likely a thermophile, rather than a hyperthermophile or mesophile [41]. Butzin et al. [44] also conducted similar ancestral sequence reconstruction experiments, and reported that the environmental temperatures of the most recent common ancestor of Thermotogales, an order of hyperthermophilic bacteria, were higher than those of its descendants, which are all hyperthermophilic.

As mentioned above, ancestral sequences of some proteins have been computationally predicted using a phylogenetic tree, and homologous amino acid sequences contained in the tree. However, protein sequences have evolved at different rates, and many mutations have accumulated during evolution. Therefore, for many proteins, it is very difficult to follow homology far back in time. Nucleoside diphosphate kinase (NDK) is distributed among Bacteria, Archaea and Eukarya, and most extant organisms possess the gene. Therefore, it is a reasonable assumption that ancient organisms also had an NDK gene. In addition, NDK sequences are well conserved among species, and a multiple alignment of extant NDK sequences contains few insertions/gaps that often interfere with the process of predicting reliable ancestral sequences. Therefore, one can suppose that an ancestral NDK sequence can be predicted with a high degree of confidence. However, a predicted ancestral sequence is also affected by the topology of the tree used to infer the sequence, and it is not possible to predict a definitively true tree topology. Indeed, while a number of phylogenetic and phylogenomic studies propose three domains in a universal tree of life [30,68,69,70,71], other studies instead support a two-domain hypothesis [72,73,74,75]. The tree illustrated in Figure 1 supports the two-domain hypothesis [18]. Therefore, three independent phylogenetic trees were built that differed in topology. Then, ancestral sequences of NDK were inferred using each tree, and experimentally reconstructed. The reconstructed proteins showed extreme thermal stability and high optimum temperature for catalytic activity, thus supporting the thermophilic ancestry of life [15]. The result is robust because similar characteristics were predicted for the ancestral proteins, even when using three topologically different phylogenetic trees.

Two other concerns are the reliability of the reconstructed ancestral amino acids, and the observed high thermal stability. Indeed, the reliability of some ancestral residue reconstruction has not been high enough, and therefore may not represent true ancestral residues (although most ancestral residues are strongly supported). We found that the predicted thermal stability of ancestral NDKs are valid, even if some residues in the reconstructed sequences do not represent the true ancestral residues [15]. Eick et al. [76] also reported that the observed characteristics of reconstructed ancestral proteins are robust to the uncertainty found in inferred sequences.

Sterner and coworkers reconstructed primordial enzyme complexes thought to be possessed by extinct species [47]. They resurrected the imidazole glycerol phosphate synthase (ImGPS) complex possessed by LUCA, and the tryptophan synthase (TS) complex possessed by the last bacterial common ancestor, and found that the two subunits (cyclase and glutaminase subunits) of the ancestral ImGPS, and the two subunits (α- and β- subunits) of the ancestral TS, were all thermostable. Moreover, it was observed that the ancestral cyclase and an extant glutaminase formed a complex structure and channeled ammonia from glutaminase to cyclase. The two ancestral subunits of TS also formed an αββα complex similar to the TS complexes from extant species. The two ancestral subunits mutually activated each other, and indole was channeled from the α subunit to the β subunit, which suggested that the sophisticated enzyme complexes responsible for substrate channeling and allosteric regulation had already been established when LUCA or the last bacterial common ancestor lived. The same research group also applied the sequence reconstruction technique to investigate the evolution of a TIM barrel protein fold [50] and identify an interface hotspot in a metabolic enzyme complex [77].

## 5. Ancestral Sequence Reconstruction Using a Non-Homogeneous Model

Some computational and empirical studies that support a theory of thermophilic ancestry used homogeneous amino acid substitution models to infer ancestral sequences [15,23,34,35,78]. The homogeneous substitution models assume that the global amino acid composition does not change among lineages and along phylogenetic trees. However, this does not accurately reflect evolution, because all sequences have different amino acid compositions. Therefore, homogeneous models are likely too simplistic to reliably infer ancestral sequences. In contrast, non-homogeneous amino acid substitution models relax this constraint by allowing different lineages to have different equilibrium compositions [22,53]. Some computational studies have focused on the environmental temperatures experienced by ancient life using a non-homogeneous evolution model [53]. Boussau et al. [26] suggested that the use of a non-homogeneous model was quite important in order to infer ancestral sequences more accurately. They reconstructed ancestral sequences of rRNAs and proteins in silico, and estimated the ancestral G + C contents of rRNA and the relative frequency of particular amino acid types. Based on their calculations, the last common archaeal and bacterial ancestors were thermophiles, but LUCA was a mesophilic or psychrophilic species [26]. Groussin et al. [27] also computed the relative frequency of each amino acid in proteins using the non-homogeneous models. According to their calculations, the last common archaeal and bacterial ancestors were likely to be thermophilic, but LUCA was likely to be mesophilic. However, it is possible that the early evolution of the amino acid repertoire [46,79,80,81,82,83,84,85] affected the frequency of each amino acid in primordial proteins. Accordingly, the thermal stability of a primordial protein—and therefore the environmental temperature of its host organism estimated from an analysis of amino acid contents—are inferential unless the stability is tested experimentally.

Using a non-homogeneous substitution model, we reanalyzed the NDK sequences that were previously used to reconstruct the ancestral NDK sequences based on a homogeneous model, and inferred additional ancestral NDK sequences [17]. The newly reconstructed ancestral NDKs also showed extremely high thermal stability, further supporting our conclusion that LUCA had a very thermally stable protein, even when ancestral sequences were inferred using a non-homogeneous substitution model. We also found that the denaturation temperature of NDK is well correlated with its host’s optimum environmental temperature [15,48]. Therefore, the thermal stability of NDK works as a molecular thermometer. Using a calibration curve based on the correlation between the denaturation temperature of NDK and the optimum environmental temperature of the host, we estimated that the environmental temperature of LUCA was 97 ± 3 °C (Figure 3).

## 6. Estimating Long-Term Change in Biosphere Temperature

Other ancestral sequence reconstruction studies expanded the timescale of targets to be reconstructed. Gaucher et al. [43] comprehensively analyzed the internal nodes of the bacterial phylogeny, and estimated the environmental temperatures of ancestral bacteria that existed 0.5–3.5 Gya. Their results suggest that the bacterial ancestor was thermophilic, and adapted later to progressively lower-temperature environments over Precambrian time. This trend is similar to a gradual cooling of the ancient ocean, as suggested by the oxygen isotope compositions of marine cherts [86,87]. A similar experiment was done by Butzin et al. [44] with proteins possessed by hyperthermophilic bacteria as the targets. They reported that the environmental temperatures of the most recent existing common ancestor of Thermotogales were higher than those of its descendants, which are all hyperthermophiles.

Groussin and Gouy [88] targeted the entire domain of Archaea. They analyzed the G + C contents of rRNAs, and the frequency of each amino acid in a set of proteins possessed by archaeal ancestors at internal nodes of the archaeal phylogeny. These data were then used to estimate environmental temperatures. The results indicated that the last common archaeal ancestor was a hyperthermophile, and extant mesophilic archaea have adapted to lower environmental temperatures over evolution.

Hart et al. [45] reported a slightly different result. The reconstructed ancestral ribonuclease H1 (RNH) suggests an unfolding temperature that was higher than that of an extant mesophilic RNH, but lower than that of an extant thermophilic RNH. They argued that the high thermal stability observed for the extant thermophilic RNH is a result of gradual adaptation to higher temperatures over time.

The high thermal stabilities of these reconstructed proteins are compatible with the high environmental temperatures of Archaean life. However, these ancestral organisms may have lived in local high-temperature environments; therefore, the results may not reflect the ambient surface temperature of ancient Earth. To overcome such a problem, Garcia et al. [48] restricted their targets to ancestral NDKs reconstructed by comparing proteins from phototrophic species that required light for growth. They reconstructed ancestral NDKs that might represent the common ancestors of cyanobacteria (oxygenic photosynthetic prokaryotes), nostocaleans (later-evolved cyanobacteria), Viridiplantae (green algae and land plants) and Embryophyta (land plants only). The ancestors of cyanobacteria, nostocaleans, Viridiplantae and Embryophyta are predicted to have existed approximately 2.7–3.1 Gya, 2.1–2.3 Gya, 0.70–0.85 Gya, and 0.44–0.46 Gya, respectively. They experimentally determined the denaturation temperatures of the reconstructed NDKs and estimated the environmental temperatures as approximately 64–82 °C (ancestor of cyanobacteria), 36–56 °C (ancestor of nostocaleans), 24–62 °C (ancestor of Viridiplantae) and 20–58 °C (ancestor of Embryophyta). The results are quite consistent to those inferred from isotope-based data [86,87], and suggest a general cooling of Earth’s surface environment over time from Archaean (~55–85 °C) to present (~15 °C) (Figure 4).

## 7. Limitations of Ancestral Inference to Estimate Temperatures of Ancient Biosphere

The characterization of the physical properties of ancestral proteins reconstructed by phylogenetic approaches has provided independent evidence of the environmental conditions of ancient Earth, and complemented the data obtained from geological and paleontological studies. Techniques to infer ancestral sequences have greatly improved in the last decade, but ancestral sequences still cannot be reconstructed with absolute certainty, and there are likely a number of inaccurate ancestral residues present in inferred sequences. An inaccurate reconstructed sequence would result in an overestimate of its thermodynamic stability, especially if the sequence was inferred by maximum parsimony or maximum likelihood [62]. Their argument was based on the observation that these two algorithms tend to adopt the amino acid that occurs most frequently among modern protein sequences as the ancestral residue. The most frequent (consensus) amino acids at a site among homologous proteins have a greater contribution to thermal stability of a protein than less frequent amino acids [89,90,91,92,93]. Ancestral inference has an inherent, undeniable tendency to converge into the consensus amino acids at many positions, and it is therefore difficult to discriminate if the observed high stability is due to the antiquity of the residue or the consensus effect; as such, the environmental temperatures inferred from the thermal stability of reconstructed ancestral proteins could be overestimated.

High temperature may not have been the only environmental parameter requiring a high stability of ancient proteins. Tawfik et al. suggested that high oxidative pressure and radiation levels, the absence of cellular osmolytes (that are prevalent in thermophiles) and/or chaperones, or low fidelity of the transcription–translation machinery might be involved in the high stability of ancient proteins [94,95].

The position of the root of a tree used to infer an ancestral sequence would also affect the results. Placement of LUCA on a tree used to reconstruct ancestral sequences would require inclusion of two or more paralogous proteins that diverged prior to the appearance of LUCA [96]. In 1989, such composite trees were reported from two independent groups using elongation factor and H^+^-ATPase, respectively [97,98]. In both studies, the position of LUCA was located within the branch connecting the archaeal and bacterial common ancestors. The universal ancestor was placed at the same position on other composite trees based on isocitrate dehydrogenase and 3-isopropylmalte dehydrogenase [55], and aminoacyl-tRNA synthase [18,99,100]. However, it has also been suggested that LUCA was located within the Bacteria domain [101,102,103,104,105,106]. Therefore, the conclusion that LUCA was thermophilic or hyperthermophilic is dependent on the hypothesis that archaea and bacteria were derived from LUCA. In addition, the incorporation of information on gene duplication, lateral gene transfer, and gene loss would improve the accuracy of inferred ancestral sequences, and therefore assumptions about the physical properties of the reconstructed protein [107]. Future work should be done to account for these evolutionary events.

## 8. Conclusions

The last two decades have seen technical improvements for inferring the ancestral nucleotide and amino acid sequences of ancestral genes and proteins. Now, these improvements provide an alternative and independent method to investigate the early evolution of life and Earth’s biosphere. However, computer-based studies to predict the environmental temperature of LUCA have provided conflicting results (Table 1). It is not evident that the correlation between the composition of genes or proteins, and the environmental temperature of the host found for present-day organisms, is applicable to ancient organisms. Therefore, it may not be appropriate to infer the environmental temperature of ancient life from a predicted base or amino acid composition.

Reconstruction of ancestral proteins suggests that LUCA possessed proteins with a thermal stability similar to, or even greater than, that of extant hyperthermophilic proteins (Table 2). The results were robust to modern protein datasets, the topology of the phylogenetic tree, and amino acid substitution models that were used to infer the ancestral sequences. The uncertainty of ancestral residues associated with the ancestral sequence reconstruction did not significantly affect the predicted thermal stability of the ancestral proteins. Therefore, LUCA is likely to have been a thermophile or hyperthermophile that thrived at a very high temperature, and its mesophilic descendants were adapted to lower temperatures as Earth’s surface environment cooled over time.

Even if LUCA was a thermophile or a hyperthermophile, it does not mean that the first life on Earth was born in a high temperature environment. Rather, our conclusion is compatible with the idea that most organisms existing at the time of LUCA became extinct, and that only LUCA that was adapted to high temperatures survived when the early Earth’s temperature drastically increased due to meteorite impacts [40].

## Figures and Tables

**Figure 1 life-07-00033-f001:**
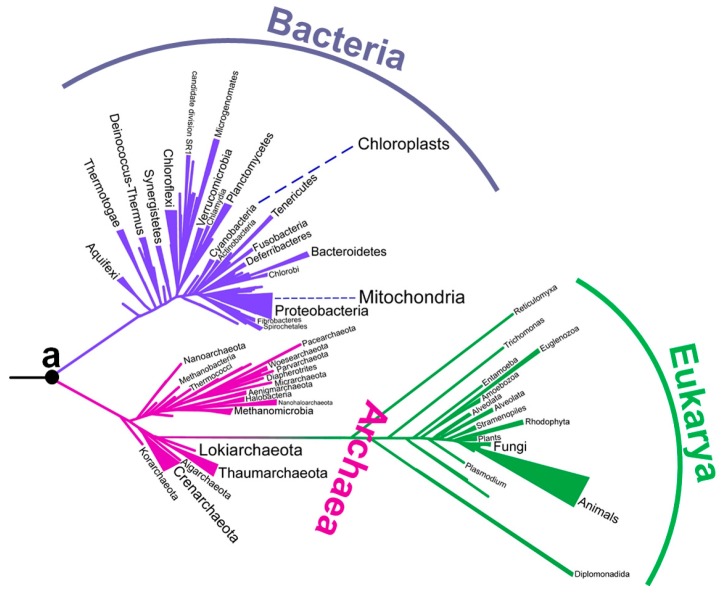
A phylogenetic tree constructed from aminoacyl-tRNA synthetase sequences [18]. The position of Commonote (LUCA) is indicated with ‘a’.

**Figure 2 life-07-00033-f002:**
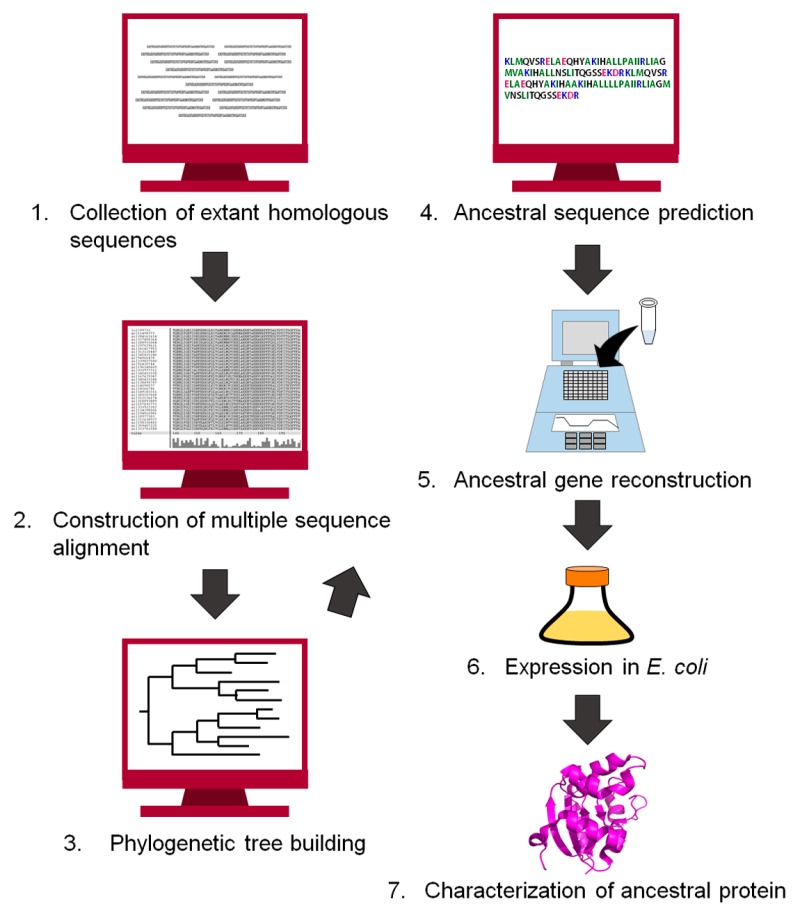
Flowchart of the procedure to infer an ancestral sequence and to reconstruct the ancestral protein in vitro.

**Figure 3 life-07-00033-f003:**
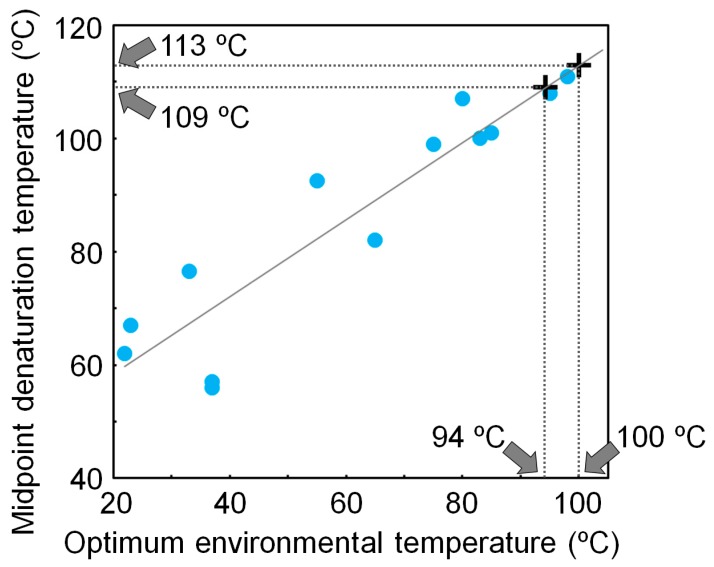
Relationship between the midpoint denaturation temperature of microbial nucleoside diphosphate kinases (NDKs) and their hosts’ optimum environmental temperatures. The optimum environmental temperature of LUCA is estimated from the calibration curve and the denaturation temperatures of the reconstructed NDKs that might be possessed by LUCA.

**Figure 4 life-07-00033-f004:**
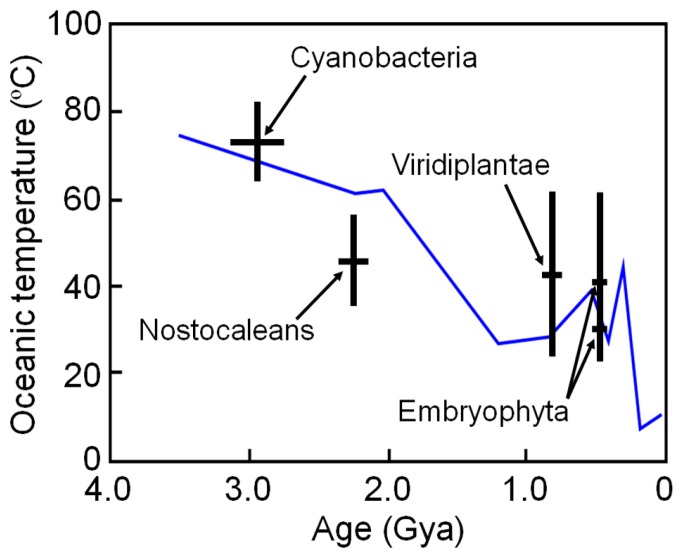
Environmental temperature ranges inferred from the midpoint denaturation temperatures of the reconstructed NDK as a function of age, with uncertainty when respective ancestral organisms first appeared. Paleotemperatures inferred from isotope-based evidence in marine cherts are also shown for comparison (blue line) [86].

**Table 1 life-07-00033-t001:** Theoretical and in silico studies for predicting the environmental temperatures of early organisms.

Target	Method	Conclusion	Refs.
Common ancestors of Archaea and Bacteria	rRNA tree	Hyperthermophilic	[19,20]
Bacterial common ancestors	rRNA tree	Mesophilic or thermophilic	[21]
LUCA	G + C content in rRNA	Mesophilic	[22]
LUCA	Reanalysis of the data used in [22]	Thermophilic or hyperthermophilic	[23]
LUCA	Evolution of reverse gyrase	Mesophilic or thermophilic	[24]
LUCA	A gene for reverse gyrase found in a gene set of LUCA	Hyperthermophilic	[25]
LUCA	G + C contents in rRNA and amino acid composition inferred using a non-homogeneous model	Psychrophilic or mesophilic	[26]
LUCA	Amino acid composition inferred using a non-homogeneous model	Mesophilic	[27]

**Table 2 life-07-00033-t002:** In vitro experimental studies for predicting the environmental temperatures of early organisms.

Target	Method	Conclusion	Refs.
LUCA	Introduction of a few amino acids into the sequence of a modern thermophilic protein	Hyperthermophilic	[55,56,57,58]
Bacterial common ancestors	Reconstruction of ancestral elongation factors	Thermophilic	[41,43]
Common ancestor of Thermotogales	Reconstruction of ancestral *Myo*-inositol-3-phospate synthases	Hyperthermophilic	[44]
LUCA	Reconstruction of ancestral NDKs using a homogeneous substitution model	Thermophilic or hyperthermophilic	[15]
LUCA	Reconstruction of ancestral NDKs using a non-homogeneous substitution model	Hyperthermophilic	[17]

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
