# Peer review of "Characterization of Reconstructed Ancestral Proteins Suggests a Change in Temperature of the Ancient Biosphere"

_life, 2017, doi:10.3390/life7030033_

Round 1

Reviewer 1 Report

The manuscript “Characterization of reconstructed ancestral proteins suggests a change in temperature of the ancient biosphere” by Satoshi Akanuma reviews work aimed at elucidating the temperature at early phases of the Earth’s biosphere. These approaches are based on ancestral sequence reconstruction (ASR) and due to different phylogenetic models used and proteins that were surrected, the ambient temperature proposed for the so-called Last Universal Common Ancestor (LUCA) varied to a great extent. The author discusses these results and presents a theory that reconciles the findings.

In general, the manuscript is well written and considers relevant literature. It was a pleasure to read it. Therefore, I have only some minor comments.

In lines 52 – 56, the author introduces a new term to name the LUCA. The author should motivate more clearly, why this is necessary and what distinguishes C. commonote from the generally accepted and widely used term LUCA. Even after having read ref [17], I did not understand the difference. In addition, the author did not consequently use this term throughout the manuscript. For example, in lines 329 and 347 - 348 he refers to the LUCA. Using two terms is puzzling and the readability would increase, if the author would use only one term (preferentially LUCA). So, I suggest, either to motivate in more detail the introduction of C. commonote and use the term consequently or use only the term LUCA.

In line 287, I would recommend to add a reference to the isotope data.

In lines 162 and 225, the same argument is repeated, although ref. [15] suggests the robustness of ASR. Please consider a rephrasing of these two sentences.

Referencing figures is inconsistent. In line 102, the term Figure 2 is used, but in lines 54 and 181, Fig. 1 (an abbreviation) is used.

Some text blocks seem to be formatted with a larger font size, like “stromatolite-like” in line 30 or “Proc. Natl. Acad. Sci” in line 371.

Author Response

Dear Reviewer

Thank you very much for reading my review manuscript and giving the positive comments.

I have incorporated all the suggestions from you into my revised manuscript. The changes that I have made in response to the comments are listed below.

Responses to Reviewer 1:

In lines 52 – 56, the author introduces a new term to name the LUCA. The author should motivate more clearly, why this is necessary and what distinguishes C. commonote from the generally accepted and widely used term LUCA. Even after having read ref [17], I did not understand the difference. In addition, the author did not consequently use this term throughout the manuscript. For example, in lines 329 and 347 - 348 he refers to the LUCA. Using two terms is puzzling and the readability would increase, if the author would use only one term (preferentially LUCA). So, I suggest, either to motivate in more detail the introduction of C. commonote and use the term consequently or use only the term LUCA.

According to the reviewer’s suggestion, I changed C. commonote to LUCA throughout the manuscript, because, as pointed out by the reviewer, the term LUCA is more commonly used than C. commonote.

In line 287, I would recommend to add a reference to the isotope data.

As suggested by the reviewer, I added the reference [86] to the isotope data in Fig. 4.

In lines 162 and 225, the same argument is repeated, although ref. [15] suggests the robustness of ASR. Please consider a rephrasing of these two sentences.

In response to the reviewer’s comment, the two sentences were removed from the revised manuscript.

Referencing figures is inconsistent. In line 102, the term Figure 2 is used, but in lines 54 and 181, Fig. 1 (an abbreviation) is used.

All figures are cited with abbreviated form in the revised manuscript.

Some text blocks seem to be formatted with a larger font size, like “stromatolite-like” in line 30 or “Proc. Natl. Acad. Sci” in line 371.

As the reviewer pointed out, some text blocks were formatted with a larger font size in the original manuscript. Therefore, I corrected the font size of those text blocks in the revised manuscript.

I think that the manuscript is improved and hope that it is now acceptable for publication in Life.

Reviewer 2 Report

This is a well written, balanced review of the field of ancestral reconstruction in relation to the thermal adaptation of Commonote. A clear overview of research in this area to date is provided with, as far as I am aware, all necessary references. I have no suggestions for changes or further comments.

Author Response

Dear Reviewer

Thank you very much for reading my review article.

I also thank you for your positive comment.

Sincerely,